# Identifying Ecosystem Service Trade-Offs and Their Response to Landscape Patterns at Different Scales in an Agricultural Basin in Central China

**Kun Li [1,2], Junchen Chen [1,2], Jingyu Lin [3], Huanyu Zhang [2], Yujing Xie [4], Zhaohua Li [2] and Ling Wang [2,*]**

[1] Hubei Provincial Key Laboratory of Regional Development and Environmental Response, Hubei University, Wuhan 430062, China
[2] Faculty of Resources and Environmental Science, Hubei University, Wuhan 430062, China
[3] School of Ecology, Environment and Resources, Guangdong University of Technology, Guangzhou 510006, China
[4] Department of Environmental Science and Engineering, Fudan University, Shanghai 200433, China
[*] Correspondence: wangling87@hubu.edu.cn

**Abstract:** Identifying relationships among multiple ecosystem services (ESs) at different scales and the factors affecting such relationships is the foundation for sustainable ecosystem management. A case study was conducted in the Sihu Lake Basin, an agricultural basin in Central China, to examine the interactions among ESs across different scales and the responses to landscape pattern changes (2000–2020). The results indicate that (1) agricultural land and wetlands were converted into construction land and gradually decreased in size; forestland and artificial channels gradually increased in size. (2) ESs had spatial heterogeneity in their strength at the grid and county scales. (3) Most relationships between ESs were synergistic at the grid and county scales, and most correlations increased as the scale increased due to landscape consistency. (4) The landscape metrics explained approximately 45.56–61.06% of the variations in ESs, and the main influencing factor was agricultural land. Our results demonstrated that the construction of rivers and channels, dense and widely distributed agricultural land and construction land were more positively correlated with increasing crop production and nitrogen export, whereas forestland exhibited a stronger contribution to increasing carbon storage and water yield. These findings explore appropriate management methods for agricultural development and ecological conservation in agricultural basins.

**Keywords:** ecosystem services; trade-offs; scale effects; landscape composition and configuration; agricultural basin

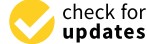



## 1. Introduction

Ecosystem services (ESs) are the benefits that people obtain directly or indirectly from the multiple processes and functions of ecosystems and are the bridge connecting the natural environment with human well-being [1]. However, approximately 60% of ESs have been degraded over the last 50 years under the pressures from global climate change and human activity, which seriously affects human well-being and threatens regional and global ecological security. Trade-off and synergistic relationships exist among multiple services provided by ecosystems, which means that the two ESs may have opposite and consistent trends [2,3]. Intensive human activities have a significant impact on the relationships between ecosystem services [4,5] and overlooking ES trade-offs and synergies may lead to a reduction in the provisioning abilities of certain ESs and may even threaten ecosystem stability [6,7]. Identifying trade-offs and synergies between ESs and clarifying their spatio-temporal variations and driving factors are crucial for optimizing multiple services and avoiding damaging other services while improving a particular service [8,9]. It is of great significance to promote ESs from theoretical research to management practice.

Numerous studies have quantified and mapped the spatial characteristics of ES trade-offs/synergies at a single point in time in highly urbanized areas, mountainous regions,

prairie areas and river basins [10–12]. Correlation analysis, scenario analysis, rose diagrams and model simulations are generally used to explore the spatial distributions, scale effects and influencing mechanisms of ecosystem service trade-offs and synergies [13–16]. These studies found that trade-off and synergistic relationships between the same pair of ESs varied in different periods and regions. The intensification of human activities (e.g., deforestation, agricultural reclamation and urbanization) can increase the supply of food and wood production services, which also causes biodiversity reduction and soil erosion, and the ecosystem services trade-off and synergy become increasingly prominent [17,18]. For instance, carbon storage has been shown to have synergy with biodiversity in rapidly urbanized cities [19]. However, there is a trade-off between them in the natural resource protection region [20]. Few studies have investigated how ES trade-offs change at different spatial scales over a long time in agricultural basins. Agricultural basins, as hydrographic basins with predominantly agricultural land use, provide different types of services (e.g., food production, water purification and biodiversity) and human well-being for regions of different scales. The contradiction among agricultural development, rapid urbanization processes and ecological protection leads to complex spatial heterogeneity in the interactions between ESs. The spatial distribution, trade-offs and synergies of ESs are influenced by temporal and spatial scales [21,22]. The scale-relevant information on the relationship among multiple ESs is effective for ES protection and management [23,24]. Most studies have examined ES trade-offs and synergies at specific scales, such as raster grids and administrative districts. Single-scale observations may capture, miss or distort interactions between ESs. It is essential for researchers to know that ES trade-offs and synergies will change over time and across spatial scales. However, how to identify ES trade-offs and synergies across time periods and combine small-scale studies with large-scale evaluations in agricultural basins is still unclear. Therefore, exploring ES trade-offs/synergies at various spatio-temporal scales is an important basis for achieving the multi-objective operation and management of ecosystems in agricultural watersheds.

Landscape patterns and human activities are considered to be the most important factors leading to scale differences in ES trade-offs. The effects of landscape patterns on ESs in lake and river basins have been examined in previous studies, and most of them have demonstrated that landscape patterns are critical to ESs [25]. Initially, studies were conducted on landscape composition to explain its impact on ES indicators [26,27]. Landscape types that are related to human development activities (e.g., urban land, agricultural land) are often negatively correlated with ESs, whereas undeveloped lands (e.g., wetland, grass land, forestland) generally have positive impacts on ESs [28]. For example, a large area of vegetation can provide a high level of regulating services (e.g., water conservation and climate regulation), but the ability to produce food will be reduced, leading to a lower level of supply services [29]. Recently, many studies paid attention to the interaction between the spatial structure of the landscape and ESs [30,31]. The landscape metrics of total landscape area, patch area and Simpson's diversity index have been analyzed to identify their relationships with ESs and have become an important approach that combines methods of landscape ecology and GIS techniques [32]. Many landscape metrics have been reported to be significantly correlated with ES trade-offs/synergies [33]. However, existing studies have not clearly distinguished how landscape composition (e.g., percentages of landscape) and landscape configuration (e.g., the shape index, density, proximity and splitting index) directly or indirectly influence the generation and use of ESs and have failed to quantitatively distinguish the difference between the two impacts on ESs. In fact, ESs are generated by different landscape types, and the random combination and spatial structure change of landscape types directly affect the supply and demand of corresponding ESs by changing ecosystem components, structures and processes [34,35]. Therefore, it is essential to explore whether landscape composition or configuration has more significant impacts on ES trade-offs/synergies to better understand the influence of landscape management and ecosystem protection in agricultural basins.

The Sihu Lake Basin (SHLB) is the largest agricultural producing area in Central China. With rapid urbanization and agricultural development in the past 20 years, agricultural planting and human activities have become the main contributors to nutrients in this area, and these changes have resulted in water quality degradation [36]. The local government is facing the challenge of balancing agricultural production, ecological protection and relevant ESs by regulating land-use patterns. Therefore, it is essential to explore ES trade-offs that vary across different scales and their responses to landscape patterns in this typical agricultural basin to optimize sustainable land-use strategies. Four research objectives were explored in this study: (1) to examine spatial and temporal changes in landscape patterns in 2000, 2010 and 2020; (2) to quantify and map trade-offs/synergies among the four ESs at grid and county scales (crop production, water yield, water purification, carbon sequestration); (3) to identify the different impacts of landscape composition and configuration on ESs; and (4) to suggest landscape management strategies for maintaining ESs to ensure sustainable development in agricultural basins.

## 2. Materials and Methods

### 2.1. Study Area

The SHLB ($112°00'$–$114°00'$ E, $29°21'$–$30°00'$ N) is located in the central Hubei Province in central China, covering an area of 11,547.5 km$^2$ (Figure 1), and agricultural land accounts for 71.4% of the total area. The basin has a subtropical monsoon climate and the elevation of the SHLB is $-54$–228 m. The surface water system is complex in the SHLB, with more than 100 rivers and 16 lakes (area > 1 km$^2$), including Honghu Lake (area = 344 km$^2$), which is the seventh largest lake and a national wetland protection area in China. This basin has rich biological resources and biodiversity, particularly agricultural vegetation and aquatic vegetation, which have great significance for economic development and ecological conservation. With fertile land and a large network of rivers and lakes, the SHLB is an important crop production base in China, containing 10 counties and 109 administrative villages. Rice, soybean, cotton and wheat are the main types of agricultural vegetation in the SHLB; large areas of these crops are grown in this basin, and the grain output accounts for more than 15% of the total output in Hubei Province. The structure and function of landscapes have been significantly influenced by the development of agriculture and urbanization, and the interaction of ESs changed during the same periods. Therefore, the effects of human activities on ecological conservation in the SHLB cannot be ignored.

### 2.2. Data Requirement and Preparation

Spatial data and statistical data were collected and analyzed in our study, as shown in Table 1. The data sets used were converted to a grid resolution of 30 m when computing the ESs.

**Table 1.** Data source and description for ecosystem services assessment.

| | Data Type | Format | Data Source |
|---|---|---|---|
| The spatial data | Land use/land cover (LULC) data | Grid size of 30 m × 30 m in 2000, 2010 and 2020 | Resources and Environmental Sciences Data Center (http://www.resdc.cn, accessed on 10 October 2021). |
| | Soil data | Grid size of 1000 m × 1000 m in 2010 and 2020 | Harmonized World Soil Database (HWSD) (http://www.fao.org, accessed on 10 October 2021). |
| | Digital elevation model (DEM) data | Grid size of 30 m × 30 m in 2010 | Tuxinggis (http://www.tuxingis.com, accessed on 10 October 2021). |
| | Normalized difference vegetation index (NDVI) data | Grid size of 1000 m × 1000 m in 2000, 2010 and 2020 | Resources and Environmental Sciences Data Center (http://www.resdc.cn, accessed on 18 January 2022). |
| | Meteorological data | Points in 2000, 2010 and 2020 | China Meteorological Data Network (https://data.cma.cn, accessed on 16 October 2021). |
| The statistical data | Crop yield, nitrogen fertilizer, water quality, population and GDP | County and township in 2000, 2010 and 2020 | Hubei Provincial Statistics Bureau, Jingzhou Statistics Bureau, Jingmen Statistics Bureau, and Qianjiang Statistics Bureau (http://tjj.hubei.gov.cn, accessed on 16 October 2021). |

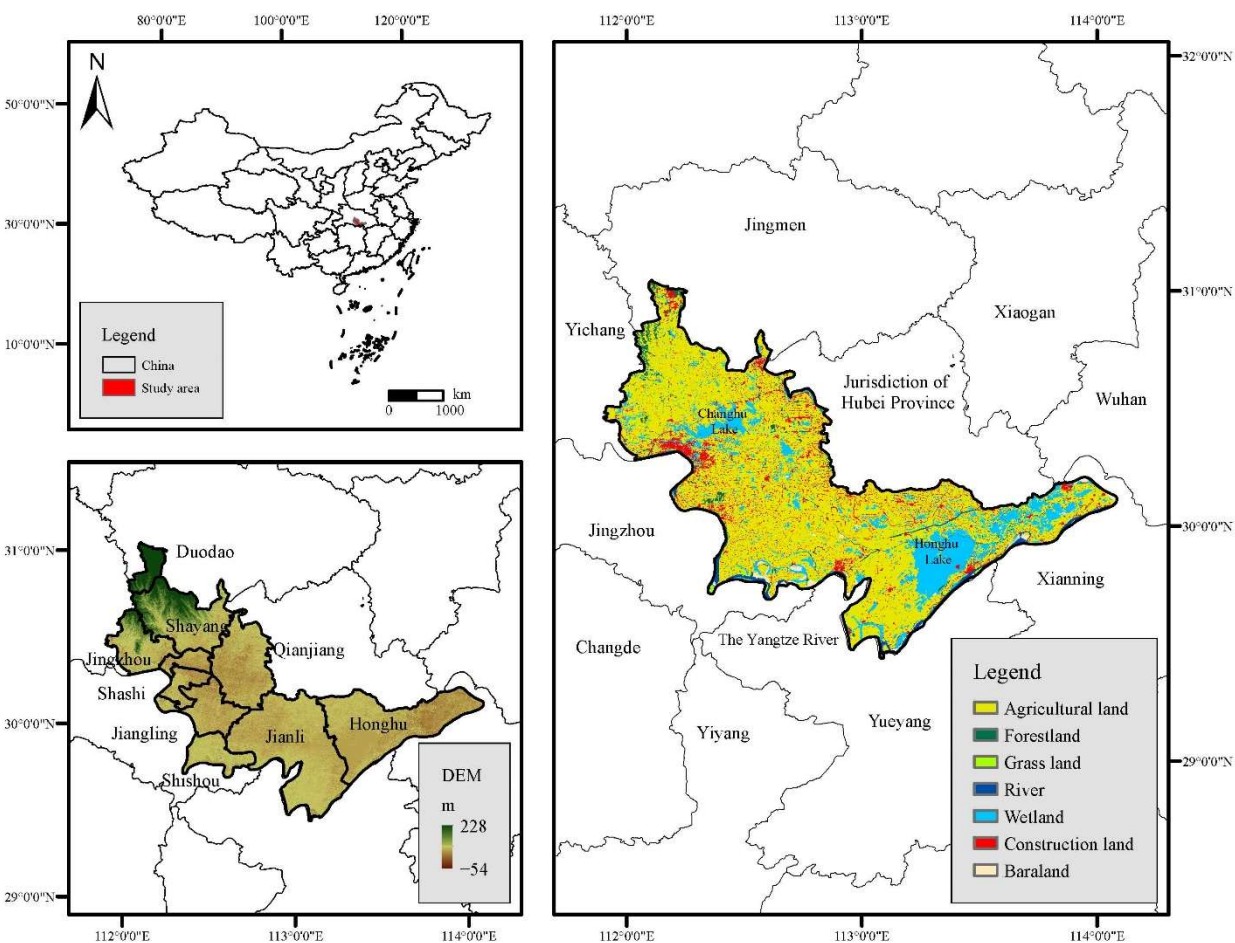

**Figure 1.** Location of the Sihu Lake Basin in central China.

### 2.3. Landscape Pattern Analyses

SHLB serves critical functions in agricultural production, flood regulation and wetland protection for local areas. Different land-use types undertake different ecosystem service functions [37], and the artificial channels and rivers were extensively modified to promote agricultural production in the study area. Therefore, LULC was classified into seven types with a grid size of 30 m × 30 m using ArcGIS 10.2 (i.e., agricultural land, forestland, grassland, wetland, construction land, rivers and bare land). To identify the effects of landscape composition and configuration on ESs, three criteria were used to select the landscape metrics for the SHLB: (1) representing important components of landscape patterns; (2) having direct or indirect relationship to the four ESs; and (3) linking to management options. Landscape composition metrics (i.e., percentage of landscape (PLAND)) and landscape configuration metrics (i.e., largest path index (LPI), patch density (PD), mean patch size (MPS), landscape shape index (LSI), edge density (ED), interspersion and juxtaposition index (IJI) and Shannon's diversity index (SHDI)) were selected as landscape characteristic metrics for the SHLB. Seven landscape metrics (i.e., PLAND, LPI, PD LSI, MPS, IJI and ED) were calculated at the class level, and the SHDI was calculated at the landscape level using FRAGSTATS 4.2 software (University of Massachusetts Amherst, Amherst, MA, USA).

### 2.4. Methods for Evaluating Ecosystem Services

The SHLB is an agricultural production area with fertile soil and rich water resources, and four ESs (e.g., crop production, nitrogen export, carbon storage and water yield) were selected with consideration of the following criteria: (1) the selected ESs should be strongly

related to agricultural production and water environment protection. (2) The selected ESs should be significantly impacted by human activities and land-use change. (3) The data needed to calculate the selected ESs should be available. Based on these criteria, we conducted ES assessments for 2000, 2010 and 2020 at the grid and county scales. The grid scale and county scale were defined as the grid resolution of 30 m and the administrative boundary in the SHLB, respectively (Table 2).

**Table 2.** Indicators used to quantify ecosystem services.

| Ecosystem Services | Abbreviation | Descriptions of Methods |
|---|---|---|
| Crop production | CP | CP service of agricultural land for each county was calculated by annual yield of grain crops in 2000, 2010 and 2020. Then, the CP of these ten counties was allocated to each grid of agricultural land according to the maximum internal NDVI of 2000, 2010 and 2020. Finally, the CP value was spatially downscaled from the county scale to the grid scale using the following formula [11,12]. $G_{ij} = NDVI_{ij}/NDVI_{mean,j} \times G_j$ where $i$ is the $i$th grid of the agricultural land layer in county $j$, $G_{ij}$ represents crop production allocated by the $i$ grid of agricultural land in county $j$, $Gj$ represents the crop production of agricultural land in county $j$, $NDVI_{ij}$ indicates the NDVI of the $i$ agricultural land grid in county $j$ and $NDVI_{mean}, j$ is the average value of agricultural land NDVI in county $j$. |
| Nitrogen export | NE | NE service was calculated by the InVEST nutrient delivery ratio model, and it was calculated with a grid resolution of 30 m using the following formula [31]. $ALV_x = HSS_x \cdot pol_x$ where $ALV_x$ and $pol_x$ represent the adjusted load value and output coefficient of grid $x$, respectively. $HSS_x$ represents the hydrological sensitivity score of the calculation method of grid $x$. |
| Carbon storage | CS | CS was calculated by the InVEST carbon storage and sequestration model with a grid resolution of 30 m using the following formula [25]. $CA_x = PA(C_A + C_B + C_S + C_D)$ where $CA_x$ represents the carbon stored in each grid $x$, $PA$ represents the grid size of 30 m $\times$ 30 m, and $C_A$, $C_B$, $C_S$ and $C_D$ indicate the density of aboveground carbon, belowground carbon, soil carbon and dead mass carbon, respectively. |
| Water yield | WY | WY was assessed by the WY module of the InVEST model and water balance equation, and it was calculated with a grid resolution of 30 m using the following formula [25,31]. $Y_x = \left[1 - \frac{AET_x}{P_x}\right] \cdot P_x$ where $Y_x$ represents the WY, $AET_x$ and $P_x$ represent the annual evapotranspiration and the annual precipitation of grid $x$, respectively. |

*2.5. Methods for Measuring Trade-Offs/Synergies and the Relationship between Landscape Patterns and Ecosystem Services*

The correlation coefficient reflects the correlation between different variables and is often used to quantify the trade-off and synergistic relationships of ecosystem services [38,39]. In this study, Pearson's correlation analysis was used to identify ES trade-offs and synergies at two scales in 2000, 2010 and 2020 using R software (version 3.6.3, R Core Team, Vienna, Austria). When the correlation coefficients of two ESs were significantly negative ($p < 0.05$) or significantly positive ($p < 0.05$), the two ESs had trade-off or synergistic relationships, respectively. Redundancy analysis was performed and images were plotted to determine the impact of landscape patterns on the ESs at different periods. The detrended correspondence analysis gradient axis was tested in CANOCO 4.5 software (Microcomputer Power, Ithaca, NY, USA) before using redundancy analysis, and the result showed that the longest gradient was below 3. Therefore, the linear model of redundancy analysis was used for the gradient analysis of the landscape pattern/ES correlations in 2000, 2010 and 2020 [40].

## 3. Results

### 3.1. Spatial and Temporal Changes in Landscape Patterns

Agricultural land, construction land and wetland were the dominant land-use types in the SHLB between 2000 and 2020, which together accounted for more than 88% of the total area (Figure 2). Four land-use types (i.e., agricultural land, wetland, grassland and bare land) decreased, and three land-use types (i.e., construction land, forestland and rivers) increased from 2000 to 2020. Agricultural land was mainly distributed over most areas of the SHLB, and the total area of agricultural land decreased from 8374.35 km$^2$ in 2000 to 8241.52 km$^2$ in 2020. This was mainly due to the conversion of agricultural land to other land-use types during the study period, with 371.62 km$^2$ of agricultural land converted into construction land and 424.01 km$^2$ converted into wetlands from 2000 to 2020 (Figure 2). The wetland area decreased from 1968.67 km$^2$ in 2000 to 1947.99 km$^2$ in 2020 due to the conversion to agricultural land and construction land. In comparison, construction land was sporadically distributed around the Yangtze River, Changhu Lake and Honghu Lake. The construction land increased from 824.99 km$^2$ in 2000 to 968.88 km$^2$ in 2020. This increase was because 41.8% of construction land was converted from agricultural land and wetlands. Moreover, there were significant spatial variations in wetlands and construction land, which decreased and became more concentrated in the SHLB, respectively. The river area also increased from 394.67 km$^2$ to 406.56 km$^2$ because channels were built for agricultural irrigation during that period. Forestland was distributed around the edges upstream, and the area slightly increased from 246.47 km$^2$ to 259.75 km$^2$ during 2000–2010, whereas it decreased to 253.17 km$^2$ in 2020. The grassland and bare land areas did not change significantly during that period, and the area accounted for 0.4% and 1.1% of the total area in 2020, respectively.

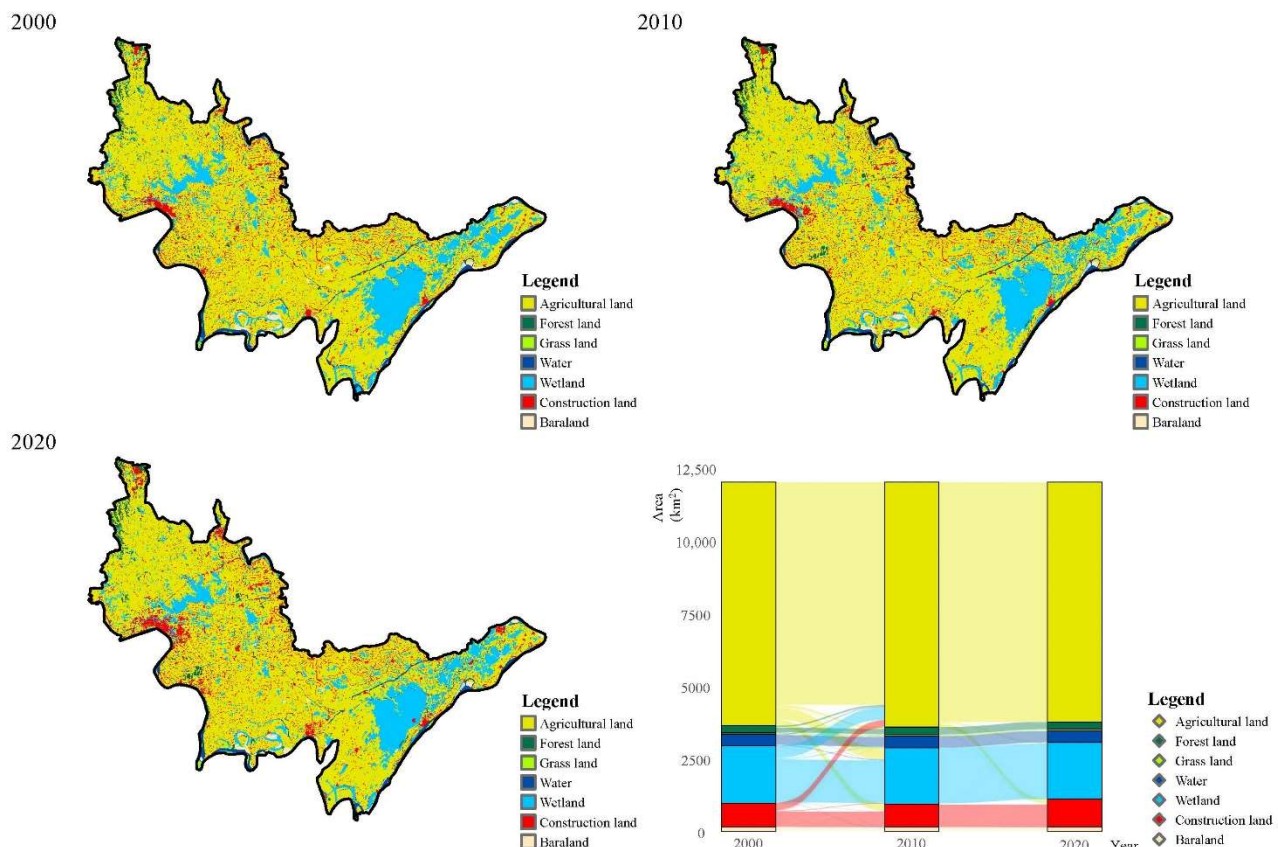

**Figure 2.** Spatial distribution and transfer of land use from 2000 to 2020.

*3.2. Changes in Ecosystem Services at Different Scales*

3.2.1. Grid Scale

The spatial distribution of ESs varied greatly from 2000 to 2020 at the grid and county scales. ESs were evaluated at the grid scale, and it was found that CP and WY displayed an increasing trend, while NE first increased and then decreased during the whole study period (2000–2020) (Figure 3). Specifically, CP, WY and NE services increased significantly by 19.5%, 252% and 0.2%, respectively, and CS decreased by 0.8% between 2000 and 2020. Spatially, the high-provision areas for CP were mostly distributed in the northern and central areas of the SHLB. The low-provision areas were mostly distributed in the east and northwest, where construction land and wetlands were distributed. The high-provision areas for NE and WY services were mainly distributed in the eastern area of the SHLB, while the high-provision areas for CS were the opposite. The spatial changes in CP and NE were widely distributed throughout the basin. However, the decline in NE was more obvious along the Yangtze River, where most urban areas were distributed. The spatial changes in WY mostly occurred in the east and south of the SHLB, and CS showed little change in most regions during the entire study period.

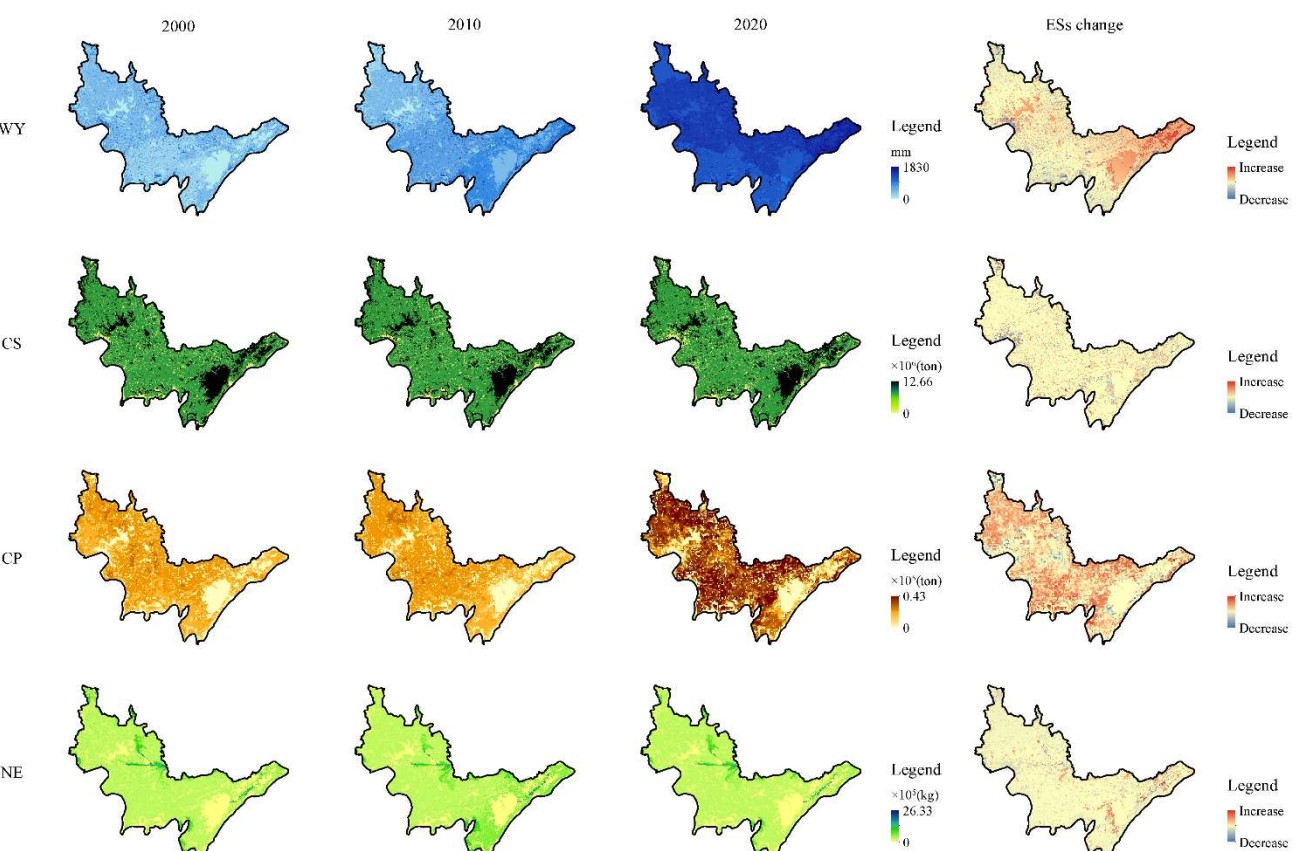

**Figure 3.** Spatial change in ecosystem services at a grid resolution of 30 m from 2000 to 2020.

3.2.2. County Scale

At the county scale, all counties showed the same change trend in these four ESs from 2000 to 2020 (Figure 4). CP, NE and CS were highest in Jianli County and Shayang County, which had rapid agricultural development. The low-provision areas for CP and NE were located in Shashi County and Jingzhou County, which had areas with high urbanization.

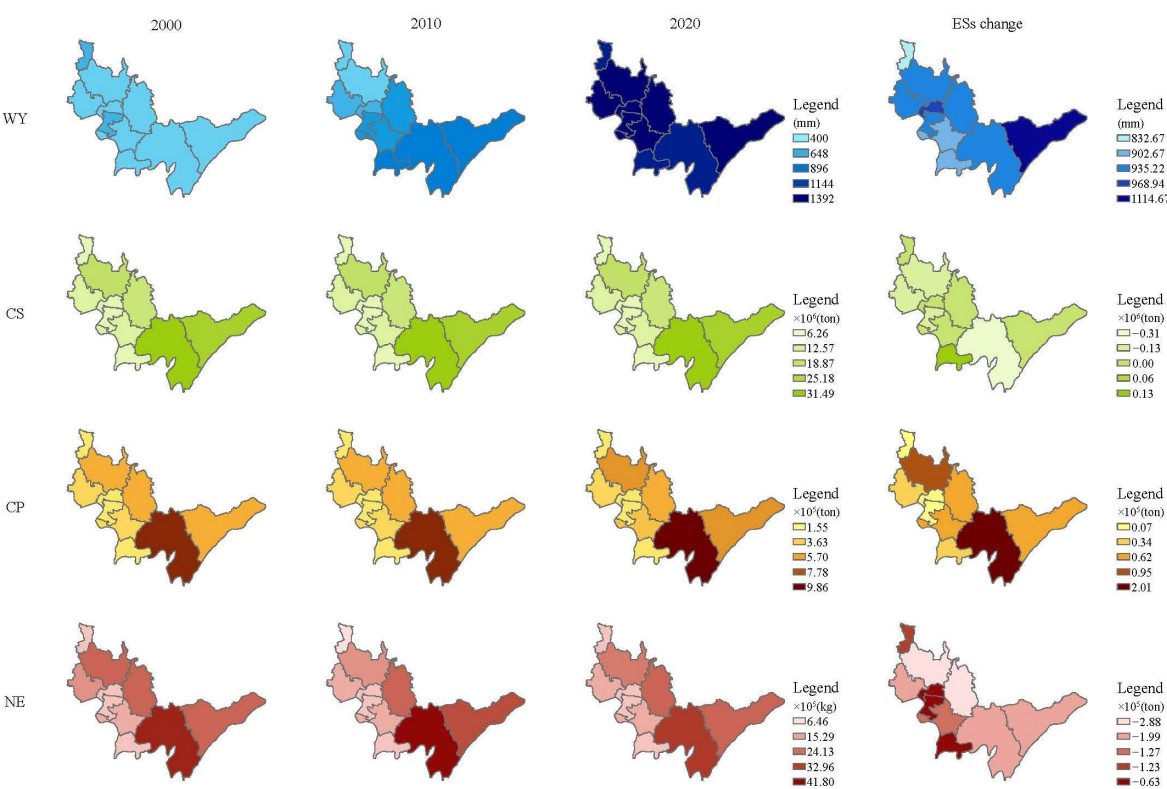

**Figure 4.** Spatial change in ecosystem services at the county scale from 2000 to 2020.

### 3.3. Trade-Offs between Ecosystem Services at Different Scales

At the grid scale, we found that all ESs showed significant ($p < 0.05$) or strongly significant ($p < 0.01$) positive or negative correlations (Figure 5). CP showed a strongly significant trade-off ($p < 0.01$) with CS services in 2000 and strongly significant synergies with WY and NE between 2000 and 2020, respectively. The correlation coefficient of CP with other services increased from 2000 to 2020. CS showed strongly significant trade-offs ($p < 0.01$) with CP, WY and NE services over time. NE services showed strongly significant synergies with CP and WY and strongly significant trade-offs with CS from 2000 to 2020. WY services showed strongly significant synergies ($p < 0.01$) with NE services and strongly significant trade-offs ($p < 0.01$) with CS services. Moreover, WY showed strongly significant synergy with CP from 2000 to 2020.

Trade-offs and synergies among the ESs varied when the ESs were assessed at the county scale. The most strongly significant ($p < 0.01$) synergies among CP, CS, WY and NE services were found from 2000 to 2020 at the county scale (Figure 5). However, there was no significant relationship between NE and the other ESs in 2020. The synergies among ESs at the county scale suggested that when CP services increased from 2000 to 2020, CS and WY increased at the same time. However, the correlations of NE with other services were not significant from 2010 to 2020. This result is consistent with our finding that CP and WY services increased from 2000 to 2020, and NE services increased from 2000 to 2010 and decreased from 2010 to 2020 over the whole basin. Moreover, the trade-offs among ESs at the county scale were not consistent with those at the grid scale. This result indicates that a decrease or increase in one ES in a grid might cause the same changes in another ES in the same grid. However, this does not suggest that the total decrease or increase in an ES in the SHLB would certainly cause the same variation in another ES.

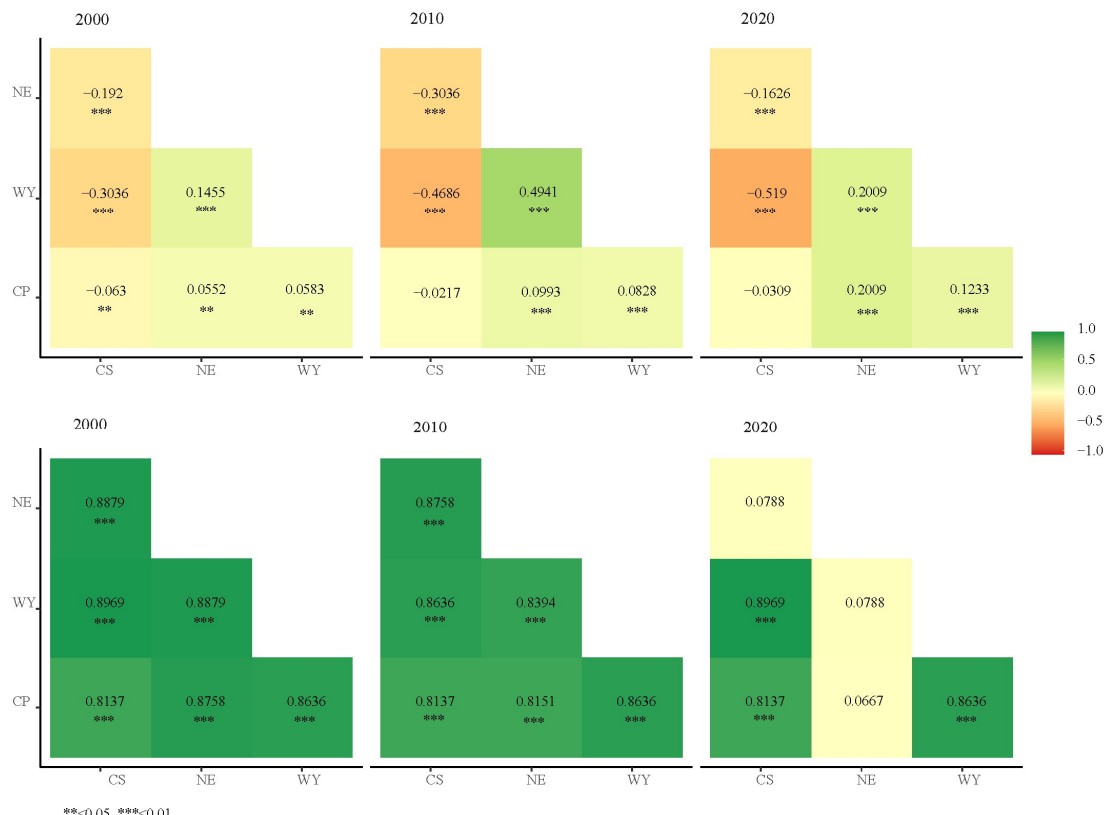

**Figure 5.** Trade-offs and synergies among ecosystem services over time at grid and county scales in the Sihu Lake Basin.

### 3.4. Relationship between Landscape Metrics and Ecosystem Service Trade-Offs

The effects of landscape composition and configuration on the four ESs demonstrated significant differences during the study period (2000–2020), as shown in Table 3 and Figure 6. The selected landscape metrics explained approximately 45.56–61.06% of the ESs, and the explanation ability increased from 2000 to 2020. This result demonstrates that the landscape pattern played a more important role in influencing ESs as the land-use types changed. In 2000 and 2010, PLAND of agricultural land, PLAND of forestland and PLAND of construction land were the most significant contributors in explaining ESs, whereas PD of agricultural land, LPI of agricultural land and PD of rivers were the main influencing factors in 2020. This result indicates that landscape composition had a greater impact on ESs than did landscape configuration in 2000 and 2010, but the impact of landscape configuration became stronger in 2020.

**Table 3.** Significant landscape metrics with the largest explanatory power for ecosystem services from 2000 to 2020.

| Year | Significant Variables | Cumulative Explained Variance (%) | | Axis 1 | Axis 2 | Axis 3 | Axis 4 | Total Explained Variance (%) |
|---|---|---|---|---|---|---|---|---|
| | PLANDagr | 21.83 | Eigen values | 0.2183 | 0.1501 | 0.0872 | 0.0383 | |
| 2000 | PLANDfor | 36.84 | CPC [1] | 44.19 | 74.59 | 92.25 | 100.0 | 45.56 |
| | PLANDcon | 45.56 | | | | | | |
| | PLANDagr | 27.44 | Eigen values | 0.2744 | 0.1432 | 0.1323 | 0.0343 | |
| 2010 | PLANDfor | 41.76 | CPC | 46.97 | 71.48 | 94.13 | 100.0 | 54.99 |
| | PDcon | 54.99 | | | | | | |
| | PDagr | 28.34 | Eigen values | 0.2834 | 0.2053 | 0.1219 | 0.0511 | |
| 2020 | LPIagr | 48.87 | CPC | 28.34 | 48.87 | 61.06 | 100.0 | 61.06 |
| | PDriv | 61.06 | | | | | | |

[1] CPC indicates the cumulative percentage correlation of landscape metrics-ecosystem services data.

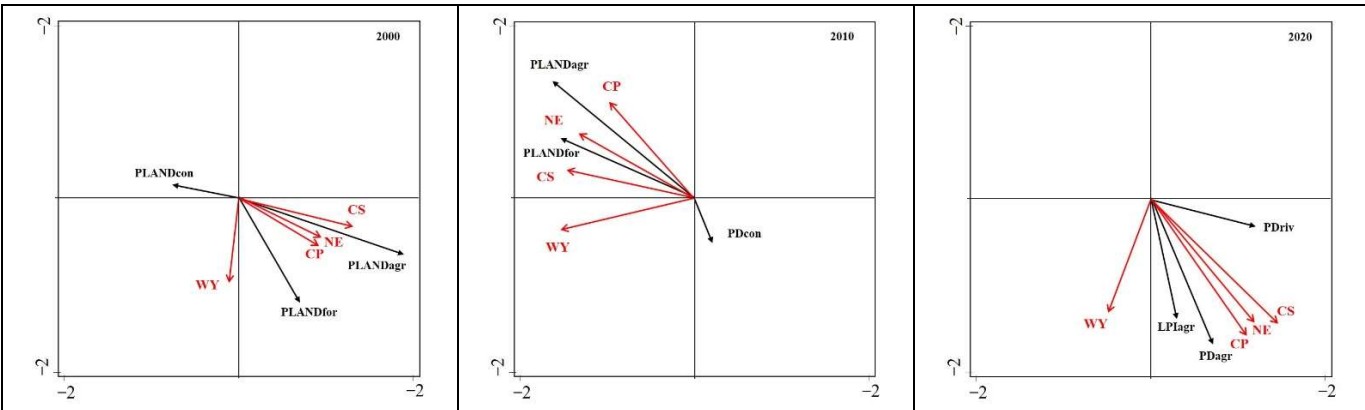

**Figure 6.** Redundancy analysis results of ecosystem services and landscape metrics from 2000 to 2020. PLANDagr: PLAND of agricultural land, PLANDfor: PLAND of forestland, PLANDcon: PLAND of construction land, PDcon: PD of construction land, PDriv: PD of rivers and LPIagr: LPI of agricultural land. CP: crop production, NE: nitrogen export, CS: carbon storage and WY: water yield. The arrows of the two variables pointing in the same direction indicate a positive correlation, and vice versa. The angle between two arrows is inversely proportional to the degree of their correlation.

More details about the impacts of the significant landscape metrics on ESs are shown in Figure 6. Generally, the PLAND of agricultural land was positively correlated with CP and NE in 2000 and 2010. However, the PD of agricultural land and LPI of agricultural land were the main landscape metrics that were positively correlated with CP and NE in 2020. The PLAND of forestland had the most significant role in explaining WY and CS in 2000 and 2010, whereas it had no positive or negative correlations with ESs in 2020. The PLAND of construction land and PD of construction land were negatively correlated with CP and NE and had no significant correlations with WY and CS. The PD of rivers was positively correlated with CP and NE in 2020. The results indicate that agricultural land had the most significant effects on ESs in the SHLB during the whole study period. Forestland and construction land were also important contributors to ESs in 2000 and 2010; however, rivers played a more important role in 2020 when land-use types varied from 2010 to 2020.

## 4. Discussion

### 4.1. Quantification of Ecosystem Services and Trade-Offs/Synergies

Exploring the relationship between different ecosystem services could help policy makers achieve an effective ecosystem conservation approach from the interchange of trade-offs/synergies among multiple services [41]. In this study, the spatio-temporal changes in the patterns and correlations of four ESs were analyzed and compared at the grid and county scales in agricultural basins. Synergy mostly occurred in CP–NE, CP–WR and NE–WR, and the relationship between CP and NE increased during the study period. This result indicates that the increase in CP was accompanied by an increase in nitrogen emissions in the SHLB. These results are consistent with a study in another agricultural basin in China, which was dominated by agricultural land with a high intensity of agricultural production and a high intensity of nitrogen provision to crops, and water quality was also polluted by agricultural planting [42,43]. However, this result is contradictory to the findings in previous studies in that agricultural production may have less effect on ESs than industrial production [44,45]. In this agricultural basin, agricultural production is an important factor in nitrogen pollution.

Generally, changes in the area of agricultural land are the most significant influencing factor leading to CP service variations [46,47]. Although the area of agricultural land decreased from 2000 to 2020, CP showed an increasing trend during these years in our study. This result implies that other important factors influenced CP, and the amount of precipitation and fertilizer applied were probably another influencing factor in the rate

of CP change [48]. The present analysis showed that the amount of N fertilizer increased from $9.25 \times 10^6$ tons to $11.49 \times 10^6$ tons between 2000 and 2020 in the SHLB. This was also supported by our finding that an increasing synergy existed between CP–NE and CP–WY from 2000 to 2020. This result is consistent with research in the agricultural area of the middle-lower Yangtze Plain on the Northeast China Plain, where with limited and decreased agricultural land, the amount of fertilizer was increased to support high crop yields [36,49]. This largely explains the increase in total CP and its synergistic relationship with NE and WY despite the decrease in agricultural land. The trade-off analysis results confirm the findings that region ecological environment should be protected while simultaneously increasing agricultural production [50]. Therefore, accurately understanding the spatial distribution and trade-offs/synergies of ESs are important for achieving ecological conservation and sustainable agricultural development, which is essential for human well-being.

However, the trade-offs between CS–CP, CS–NE and CS–WY were revealed in our results and differed from past research in the Taihu Lake Basin [51]. This is because CS in the basin was mainly distributed in forest areas and wetlands with higher slopes, where CP, NE and WY were low. Moreover, Hou et al. [52] and Peng et al. [43] found a trade-off between WY and NE, while the relationship was synergistic in our study. This difference was explained by the fact that WY and NE were distributed in agricultural areas, and the spatial pattern of precipitation and evaporation positively contributed to increases in WY and NE, which led to similar spatial distributions and synergies of WY and NE. The results reveal that the changes in different land-use types were the main impact factors causing trade-offs and synergies among ESs, which should be considered carefully in regional land-use management.

### 4.2. Scale Effect of Trade-Offs/Synergies

Most studies suggest that ES trade-offs and synergies may vary with scale [22,53,54]. Thus, it is critical to distinguish the relationships between multiple ESs across different spatio-temporal scales. In our study, we found significant synergies between WY–NE, CP–WY and NE–CP over time and significant trade-offs between CS–WR, CS–CP and CS–NE at the grid scale. However, the relationships between WY–NE, CP–WY and NE–CP varied to strongly significant synergies, and the relationships between CS–WR, CS–CP and CS–NE changed from trade-offs to strongly significantly synergies at the county scale. This result indicates that the scale not only impacted the direction of interactions between CS–WR, CS–CP and CS–NE but also impacted the significance of the interactions between WY–NE, CP–WY and NE–WY. This was consistent with the finding of a previous study by Yang et al. (2021) [55], who concluded this result to land-use consistency; that is, a certain type of land-use can affect two or more ESs simultaneously. Although most of the correlations between the four ESs in the SHLB were significantly synergistic at the grid and county scales, there are still some differences between our results and those of other studies examining agricultural watersheds. Moreover, our results indicate that the trade-offs/synergies at the grid scale could not represent the same relationship that existed at the county scale. The main reason is that when grid units are aggregated to a broad scale, the composition and configuration of different landscape patterns change, which determines the ESs of the counties [52,56].

### 4.3. Different Roles of Landscape Composition and Landscape Configuration in Ecosystem Services

ESs are affected by changes in the size, density, shape and diversity of landscape patterns [57,58]. Effective landscape planning and optimization require a full understanding of the effects of landscape metrics on ESs, which are still rarely studied. Our results indicate that both landscape composition and configuration had significant effects on ESs at different periods. The effect of landscape configuration on ESs has been suggested to be more obvious in previous studies [53,59]. In our study, most ESs were significantly positively or negatively correlated with PLAND in 2000 and 2010 (Figure 6), and landscape

composition seemed to have more obvious effects on ESs. However, the PD and LPI became the dominant landscape metrics in 2020. The results demonstrate that landscape composition and configuration played different roles in influencing ESs. To some extent, this result shows that the effects of landscape composition and configuration on ESs are not consistent in different places and are associated with changes in landscape patterns in the study area.

Generally, agricultural land and construction land are closely related to CP and NE in agricultural basins [60,61]. In our study, although agricultural land decreased from 2000 to 2020, it exerted a large impact on ESs. More specifically, PLAND of agricultural land, PD of agricultural land and LPI of agricultural land were mostly positively correlated with CP and NE, which implied that concentrated and extensive distributions of large, cultivated areas would lead to crop growth and water quality degradation, as suggested by previous studies [36,62]. Interestingly, the PD of construction land and PLAND of construction land were significantly negatively correlated with CP and NE in 2000 and 2010. However, these had nonsignificant negative correlations with CP and NE in 2020, which was opposite to the results reported by previous studies [63,64]. This situation may be related to the construction of municipal wastewater treatment plants in construction areas. As the sewage treatment rate increased from 32% in 2000 to 57% and 90% in 2010 and 2020, respectively, the discharge of pollutants gradually decreased in the SHLB, even when construction land increased in this period. In contrast, many studies have demonstrated that forestland is mostly related to NE and CS in different watersheds. In this study, although forestland had a small coverage area, it exerted a disproportionately large influence on the ESs. We found that the PLAND of forestland was the main landscape metric, which was significantly positively correlated with CP, NE, WY and CS. This result is consistent with other studies [65,66]. Forestland was mostly considered to have a water conservation function and reduce pollutant emissions, and it could also improve regional CS ability. Thus, we believe that a large area of forestland might decrease pollutant emissions and increase CS services and WY services more than dispersed forest landscapes in agricultural basins. It is interesting to note that rivers were positively correlated with CP and NE, which has rarely been reported in other studies. This relationship might be explained by the fact that many channels have been built for irrigation in recent years, and CP has increased, but pollutants will accumulate along channels and river networks, and affect the water environment of the SHLB.

### 4.4. Limitations and Future Research Directions

In this study, ES trade-offs/synergies were quantified at raster grid and county scales, which is an important part of research on ES trade-offs. The results are meaningful for proposing methods for coordinated agricultural and ecological development in different agricultural basins, and provide support for the sustainable and continuous development of the SHLB. Evaluating ESs in the raster grid is effective in identifying the spatial distribution of ESs in the study area. However, land-use policies and ecosystem protection are usually implemented at the administrative scale levels (e.g., county, city and province) [67]. In this study, the county scale is the most practical scale for policy-making and decision-makers who are concerned with ESs in agricultural basins. Therefore, to protect and manage ecosystems more effectively, larger or smaller administrative scales can be supplemented to explore ES changes and their response to landscape patterns in future research. This study also has limitations in the consideration of some important ESs (e.g., air purification, biodiversity soil erosion, etc.) due to poor data availability. Although the selected ES indicators depended on the ecosystems for their provision and were relevant to the agricultural basin, the availability and quality of the data severely limited the set of ESs that we could use. For optimal land-use and ecosystem management, more important functions should be selected to further assess the ESs [68]. Finally, our study demonstrated that it is essential to analyze long-term land-use changes in agricultural basins to gather more information on the multiple relationships of ESs. Future land-use planning and policies in agricultural

basins should take into consideration the importance of ES trade-offs and synergies, as well as support decision making for farmers and land managers.

## 5. Conclusions

In the SHLB, where rapid agricultural development is occurring, the composition and configuration of the landscape experienced major changes from 2000 to 2020, which had different effects on the four ESs. The results suggest that (1) agricultural land, construction land and wetland changed the most dramatically due to rapid agricultural development and urbanization in the SHLB. The responses of ecosystem services to land-use change were different; the services of CP, WY and NE showed an upward trend while CS decreased over time. The trade-offs/synergistic relationships among multiple ESs indicated that CP, WY and NE were mostly distributed and increased in agricultural areas, which was accompanied by a decrease in CS, and CP was an important factor affecting nitrogen pollution in the study area. Although agricultural land was decreasing, large amounts of fertilization were the main influencing factor for promoting crop yield and nutrient discharge. (2) The impact of landscape composition and configuration on ESs changed from 2000 to 2020, and the spatial scale impacted the direction and significance of ESs interactions when grid units were aggregated to counties. It is important for policy-makers to take different land-use management measures at different spatial scales for ecosystem protection. (3) The large areas of densely distributed agricultural land and construction land (i.e., PLAND of agricultural land, LPI of agricultural land, PLAND of construction land and PD of construction land) were identified as the most significant influencing factors that explained the variations in CP and NE and were positively correlated with increasing CP and water quality deterioration. However, the construction of municipal wastewater treatment plants is important in reducing pollutant emissions. Moreover, a large area of forestland contributed to increasing CS and WY services and decreasing nonpoint source pollution. In addition, the construction of rivers and channels significantly promoted CP services but also increased the migration and diffusion of pollutants in the SHLB. These findings provide an integrated ES trade-off assessment framework at different scales for managers to clarify how ESs are impacted by landscape composition and configuration, and the results offer important implications for reducing ecological risk from agricultural development in the SHLB as well as in similar areas in China.

**Author Contributions:** Conceptualization, K.L. and L.W.; methodology, K.L. and J.C.; software, J.C.; validation, K.L., J.L. and Y.X.; formal analysis, H.Z.; data curation, Y.X.; writing—original draft preparation, K.L.; writing—review and editing, L.W. and Z.L.; visualization, J.C.; funding acquisition, K.L. All authors have read and agreed to the published version of the manuscript.

**Funding:** This research was funded by Natural Science Foundation of Hubei Province, grant number 2021CFB045.

**Institutional Review Board Statement:** Not applicable.

**Informed Consent Statement:** Not applicable.

**Data Availability Statement:** Data is contained within the article. For detailed information of each part, please contact the corresponding author.

**Conflicts of Interest:** The authors declare no conflict of interest.

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
