# Peer review of "Identifying Ecosystem Service Trade-Offs and Their Response to Landscape Patterns at Different Scales in an Agricultural Basin in Central China"

_land, doi:10.3390/land11081336_

Round 1

Reviewer 1 Report

The manuscript titled " Identifying ecosystem service trade-offs and their response to landscape patterns at different scales in an agricultural basin in Central China " intends to explore whether landscape composition or configuration has a more significant impacts on ES trade-offs/synergies to better understand the influence of landscape management and ecosystem protection in agricultural basins. The manuscript select data from Sihu Lake Basin (SHLB) is the largest agricultural producing area in Central China.

The research is original; it could be characterized as novel and in my opinion important to the field, it also has an almost appropriate structure and the language has been used well. In the meanwhile, the manuscript has I think a small extent (about 4774 words) and it is quite comprehensive. The tables (3) and figures (6) make the paper to reflect well to the reader. For this reason, paper has a "diversity look", not only tables, not only numbers, not only words.

The title, I think, is all right. The abstract did not reflect well the findings of this study, and it has not the appropriate length. Please revise the abstract of the manuscript and do not forget abstract need to encourage readers to download the paper. The Abstract needs further work. It is not clear. Abstracts should indicate the research problem/purpose of the research, provide some indication of the design/methodology/approach taken, the findings of the research and its originality/value in terms of its contribution to the international literature. The abstract has a long length (about 251 words). Please, revise the abstract, it must be up to 200 words long, for this reason I would be good to reduce [see: Instructions for Authors / Manuscript Submission Overview / Accepted File Formats - (https://www.mdpi.com/journal/land/instructions#submission or https://www.mdpi.com/files/word-templates/land-template.dot)].

The introduction is effective, clear, and well organized; it really introduced and put into perspective what research is negotiating but is too short. Please revise the Introduction of the manuscript and include references which are already exists in bibliography (as you did). Moreover, it does not contain a clear formulation and description of the research problem. Please insert a clear description and justification of the problem the article deals with. Your literature research should be critical and more informed, rather than listing previous research. This section requires significant improvement. The methodology followed is not sufficiently documented and needs to be explained clearly.

For the Methodology chapter, the research conduct has been tested in several areas of the world, with similar results and will probably be tested in others. Appropriate references to the methodology included in the already published bibliography but you can put more references, from all the world. Do not forget, the journal “Land” is international.

The results section is good. The argument flows and is reinforced through the justification of the way elements are interpreted. But the same do not applies to the Discussion and Conclusion. It is advised to revise the Discussion and Conclusion. Both sections should be consistent in terms of Proposal, Problem statement, Results, and of course, future work. Your conclusion section is short and does not do justice to your work. Make it your key contributions, arguments, and findings clearer. You must refer to the literature and previous studies in your discussion and conclusion sections. It is recommended to remove phrases from Results and put them in the Conclusion and Discussion with nice order and to be enriched. I think all the manuscript should be increased about 1200 words including the increase in the introduction.

I would be much more satisfied if the number of references was slightly higher (about 20 - 25 references) and I would appreciate it if it also included data other than Asia for example America, Europe or Australia, because it has many references from Asia. In this way it is documented that a method that is tested in a place with its own characteristics can be implemented in other places around the world.

More discussion is needed, comparing the results of this work related to attributes with those of other studies. I believe that the conclusions section or discussion should also include the main limitations of this study and incorporate possible policy implications. I think, something more should be said about practical implications.

Please revise the references of the manuscript and include references which are already exists in bibliography. References must have an appropriate style, for this reason I would be good to change [see: Instructions for Authors / Manuscript Preparation / Back Matter / References: - (https://www.mdpi.com/journal/land/instructions or https://www.mdpi.com/authors/references)]. Do not forget, DOI numbers (Digital Object Identifier) are not mandatory but highly encouraged and make the review easier.

For example, for reference 4 you write “Berry, Z.C.; Jones, K.W.; Aguilar, L.R.G.; Congalton, R.G.; Holwerda, F.; Kolka, R.; Looker, N.; Ramirez, S.M.L.; Manson, R.; Mayer, A.; et al. Evaluating ecosystem service trade-offs along a land-use intensification gradient in central Veracruz, Mexico. Ecosystem Services 2020, 45, doi:10.1016/j.ecoser.2020.101181”. I think must be revised as “Carter Berry, Z.; Jones, K.W.; Gomez Aguilar, L.R.; Congalton, R.G.; Holwerda, F.; Kolka, R.; Looker, N.; Lopez Ramirez, S.M.; Manson, R.; Mayer, A.; et al. Evaluating Ecosystem Service Trade-Offs along a Land-Use Intensification Gradient in Central Veracruz, Mexico. Ecosyst. Serv. 2020, 45, 101181, doi:https://doi.org/10.1016/j.ecoser.2020.101181”.

Author Response

Dear reviewer,

We are very grateful and thankful to your valuable evaluation and constructive advices for our manuscript. The comments are very helpful for improving our paper and provide important guidance to our research. We have studied comments carefully and have responded one by one, which we hope meet your approval. We appreciate for your warm work earnestly, and hope that the correction will meet with approval. We sincerely hope this manuscript will be finally acceptable to be published on Land.

Point 1: The manuscript titled "Identifying ecosystem service trade-offs and their response to landscape patterns at different scales in an agricultural basin in Central China " intends to explore whether landscape composition or configuration has a more significant impacts on ES trade-offs/synergies to better understand the influence of landscape management and ecosystem protection in agricultural basins. The manuscript select data from Sihu Lake Basin (SHLB) is the largest agricultural producing area in Central China.

The research is original; it could be characterized as novel and in my opinion important to the field, it also has an almost appropriate structure and the language has been used well. In the meanwhile, the manuscript has I think a small extent (about 4774 words) and it is quite comprehensive. The tables (3) and figures (6) make the paper to reflect well to the reader. For this reason, paper has a "diversity look", not only tables, not only numbers, not only words.

The title, I think, is all right. The abstract did not reflect well the findings of this study, and it has not the appropriate length. Please revise the abstract of the manuscript and do not forget abstract need to encourage readers to download the paper. The Abstract needs further work. It is not clear. Abstracts should indicate the research problem/purpose of the research, provide some indication of the design/methodology/approach taken, the findings of the research and its originality/value in terms of its contribution to the international literature. The abstract has a long length (about 251 words). Please, revise the abstract, it must be up to 200 words long, for this reason I would be good to reduce [see: Instructions for Authors / Manuscript Submission Overview / Accepted File Formats (https://www.mdpi.com/journal/land/instructions#submission or https://www.mdpi.com/files/word-templates/land-template.dot)].

Response 1: This valuable suggestion has been well taken, and have reduced the abstract from 251 words to 199 words according to the above website.

Point 2: The introduction is effective, clear, and well organized; it really introduced and put into perspective what research is negotiating but is too short. Please revise the Introduction of the manuscript and include references which are already exists in bibliography (as you did). Moreover, it does not contain a clear formulation and description of the research problem. Please insert a clear description and justification of the problem the article deals with. Your literature research should be critical and more informed, rather than listing previous research. This section requires significant improvement. The methodology followed is not sufficiently documented and needs to be explained clearly.

Response 2: We totally agree with this suggestion, and have rewritten introduction according to the reviewer’s suggestion (lines 46-47, 53-55, 57-59, 62-63, 68-71 81-83).

Point 3: For the Methodology chapter, the research conduct has been tested in several areas of the world, with similar results and will probably be tested in others. Appropriate references to the methodology included in the already published bibliography but you can put more references, from all the world. Do not forget, the journal “Land” is international.

Response 3: Thank you very much for this useful suggestion. We have been added some references from America, Europe, Australia and other place in methodology chapter of the revised manuscript.

Point 4: The results section is good. The argument flows and is reinforced through the justification of the way elements are interpreted. But the same do not applies to the Discussion and Conclusion. It is advised to revise the Discussion and Conclusion. Both sections should be consistent in terms of Proposal, Problem statement, Results, and of course, future work. Your conclusion section is short and does not do justice to your work. Make it your key contributions, arguments, and findings clearer. You must refer to the literature and previous studies in your discussion and conclusion sections. It is recommended to remove phrases from Results and put them in the Conclusion and Discussion with nice order and to be enriched. I think all the manuscript should be increased about 1200 words including the increase in the introduction.

Response 4: We totally agree with the reviewer’s point of view. More details, such as the implications, limitation and future work have been provided in Section 4.4 (lines: 326-342), the key contributions, arguments, and findings have been added in the conclusion and discussion of revised manuscript (lines: 347-348, 352-353, 355-356, 360-361, 362-363).

4.4 Limitations and future research directions

In this study, ES trade-offs/synergies were quantified at raster grid and county scales, which is an important part of research on ES trade-offs. The results are meaningful for proposing methods for coordinated agricultural and ecological development in different agricultural basins, and provide support for the sustainable and continuous development of the SHLB. Evaluating ESs in the raster grid is effective in identifying the spatial distribution of ESs in the study area. However, land-use policies and ecosystem protection are usually implemented at the administrative scale levels (e.g., county, city, and province) [67]. In this study, the county scale is the most practical scale for policy-making and decision-makers who are concerned with ESs in agricultural basins. Therefore, to protect and manage ecosystems more effectively, larger or smaller administrative scales can be supplemented to explore ES changes and their response to landscape patterns in future research. This study also has limitations in consideration of some important ESs (e.g., air purification, biodiversity, and soil erosion, etc.) due to poor data availability. Although the selected ES indicators depended on the ecosystems for their provision and were relevant to the agricultural basin, the availability and quality of the data severely limited the set of ESs that we could use. For optimal land-use and ecosystem management, more important functions should be selected to further assess the ESs [68]. Finally, our study demonstrated that it is essential to analyze long-term land-use changes in agricultural basin to gather more information on the multiple relationships of ESs. Future land-use planning and policies in agricultural basins should take into consideration the importance of ES trade-offs and synergies, and support decision making for farmers and land managers.

Point 5: I would be much more satisfied if the number of references was slightly higher (about 20 - 25 references) and I would appreciate it if it also included data other than Asia for example America, Europe or Australia, because it has many references from Asia. In this way it is documented that a method that is tested in a place with its own characteristics can be implemented in other places around the world.

Response 5: This is a valuable comment. We have been added 22 references from America, Europe, Australia and other place in the revised manuscript.

Point 6: More discussion is needed, comparing the results of this work related to attributes with those of other studies. I believe that the conclusions section or discussion should also include the main limitations of this study and incorporate possible policy implications. I think, something more should be said about practical implications.

Response 6: We fully agree with this suggestion, and the main limitations, policy implications and related modifications have been included in Section 4.1 (lines:266-269) and Section 4.4 (lines: lines: 326-342).

Point 7: Please revise the references of the manuscript and include references which are already exists in bibliography. References must have an appropriate style, for this reason I would be good to change [see: Instructions for Authors / Manuscript Preparation / Back Matter / References: - (https://www.mdpi.com/journal/land/instructions or https://www.mdpi.com/authors/references)]. Do not forget, DOI numbers (Digital Object Identifier) are not mandatory but highly encouraged and make the review easier.

For example, for reference 4 you write “Berry, Z.C.; Jones, K.W.; Aguilar, L.R.G.; Congalton, R.G.; Holwerda, F.; Kolka, R.; Looker, N.; Ramirez, S.M.L.; Manson, R.; Mayer, A.; et al. Evaluating ecosystem service trade-offs along a land-use intensification gradient in central Veracruz, Mexico. Ecosystem Services 202045, doi:10.1016/j.ecoser.2020.101181”. I think must be revised as “Carter Berry, Z.; Jones, K.W.; Gomez Aguilar, L.R.; Congalton, R.G.; Holwerda, F.; Kolka, R.; Looker, N.; Lopez Ramirez, S.M.; Manson, R.; Mayer, A.; et al. Evaluating Ecosystem Service Trade-Offs along a Land-Use Intensification Gradient in Central Veracruz, Mexico. Ecosyst. Serv. 2020, 45, 101181, doi: https://doi.org/10.1016/ j.ecoser.2020.101181”.

Response 7: The references of the manuscript have been checked and revised one by one according to journal format of MDPI.

Reviewer 2 Report

After having read this manuscript in detail, I suggest that the following (minor) corrections are made:

i. Relationship/s (line 18 - page 1). A first of many examples of missue of singular/plurar tenses  along the text. Revise in depth and correct.

ii. What are agricultural basins?. It is either agricultural aras or hydrographic/hydrologic basins. OR hydrographic basins with predominantly agricultural land-use. Is this the case?. PleAuthors ase do provide a clear definition and/or justification. 

iii. "Grid" (I guess you mean a raster grid) to refer to scale is a confusing concept. Especially to compare with a real administrative scale level, like the county. What implications does a grid have for decision-making and administrative purposses? (I can anticipate that; none). 

iv. Positively associated or correlated?. It is either a causal relationship or a statistical one, but associations do not seem too relevant in such a context.

v. If you only examined a basin (Sihu Lake), how can you affirm that "The impacts of landscape configuration and composition on ESs varied by time and region"?, what regions have you compared?, also, what do you hereby mean by "time"?, what kind of timescale are you discussing?

vi. A deep review of English language and expression is required to avoid errors, such as "Trade-offs and synergies relationships were existed" (line 44)

vii. Some details in figure 1 are too small, and may be harder to read in printed versions of the paper.

viii. What do you mean by "unused land"?...nature dominated land?. Look for a better term. 

ix. The methods chosen in table 2 and section 2.5 could be better justified, and not merely described. Otherwise, their use may seem too random.

x. The graphic (not mapped) components in figure 2 are rather unclear

xi. Same is the case for figure 3. Also, by grid scale, what level of grid resolution do you mean?. As you know scale and level of representation may have a strong influence on results, and thus it is very relevant to be as precise as possible in this case.

xii. Figure 6 is rather difficult to read and interpret. Improve.

xiii. Along the text, relationships, correlations and trade-offs (between ES and land-use classes as well) are used apparently interchangeably, resulting in a certain confusion. This should be corrected.

xiv. Synergy vs Synergies (line 218, section 4.1). Multiple small errors like this one are found along the whole text. It would be essential if you could revise them carefully, and correct as required. Same (for example) with trade-off/trade-offs in L 235 of the following page.

xv. One example of a really confusing and badly written sentence is the following one: "In our study, we found a nonsignificant or significant synergies between WY-NE, CP-WY, and NE-CP over time and a significant or nonsignificant trade-offs between CS-WR, CS-CP, and CS-NE at the grid scale". In my view these types of inteligible statements potentially ruin the presentation of a series of data and findings that are otherwise relevant for publication. 

xvi. Last, I would suggest you revise your use of the landscape-scale outreach of your findings. Despite of indicating of a centrality to landscape configuration and related functions and processes, your study was conducted at 2 scales (administrative unit + grid), none of which correspods to the landscape. This is is a gap that should be corrected along the text.

Once these issues are proprly answered and resolved, I feel your paper could be ready for publication in Land.

Best regards

The Reviewer

Reviewer 3 Report

Identifying relationship among multiple ecosystem services at different scales and the factors affecting such relationships is the foundation for sustainable ecosystem management in agricultural basins. However, despite numerous studies conducted, there is no single point of view on indicators of ecosystem services. In addition, the problem of finding compromises between the preservation of ecosystem services and the economic development of regions is extremely urgent. From this point of view, this paper is relevant, as it draws attention to an important problem.

The research results have both theoretical and applied significance. These findings are important to clearly understand the complex interactions among ecosystem services at different scales and explore appropriate management methods for agricultural development and ecological conservation in agricultural basins.

The study area is described satisfactory. The research was conducted in Sihu Lake Basin (112°00′ - 114°00′E, 29°21′ - 30°00′N) (located in central Hubei Province in Central China). Unfortunately, the natural vegetation is not described. It is advisable to fix this. There is not enough information about the types of vegetation, species diversity, prevailing species, as well as the significance of it for humans.

The methodology is described in detail. The authors used modern methods of analysis. In this study, land cover was classified into seven types with a grid size of 30 m × 30 m using ArcGIS 10.2 ((i.e., agricultural land, forestland, grassland, wetland, construction land, rivers and unused land). It is necessary to explain the choice of classification. Pearson's correlation analysis was used to identify ecosystem services trade-offs and synergies at two scales in 2000, 2010 and 2020 using R software. Redundancy analysis (RDA) was used to identify the impact of landscape patterns on the ecosystem services at different periods. The detrended correspondence analysis was tested in CANOCO 4.5. The selected methods correspond to the objectives set.

The research results are perfectly illustrated with figures and tables that are informative and do not duplicate each other. The paper contains 3 informative tables and 6 visual figures. The results are presented clearly and clearly.

Conclusions follow from the results and are reasonable.  It is desirable to expand the description of practical application for politicians and managers. How to preserve and enhance ecosystem services of different types of lands? What should you pay attention to in the first place? What difficulties can you face when organizing sustainable land use? What uncertainties stand in the way of solving problems? The article will be of interest to a wide range of readers whose scientific interests are related to ecology. Despite the fact that English is not my native language, I read the paper with interest and had no difficulties in understanding. The paper fully corresponds to the subject and level of the Land.

The paper contains many abbreviations. This complicates the perception of the results.

Author Response

Dear reviewer,

We are very grateful and thankful to your valuable evaluation and constructive advices for our manuscript. The comments are very helpful for improving our paper and provide important guidance to our research. We paid special attention to each comment, and have responded to each one below. The corresponding changes to the text are highlighted in the revised version of the manuscript. We sincerely hope that the revision will meet with your approval.

Point 1: Identifying relationship among multiple ecosystem services at different scales and the factors affecting such relationships is the foundation for sustainable ecosystem management in agricultural basins. However, despite numerous studies conducted, there is no single point of view on indicators of ecosystem services. In addition, the problem of finding compromises between the preservation of ecosystem services and the economic development of regions is extremely urgent. From this point of view, this paper is relevant, as it draws attention to an important problem.

The research results have both theoretical and applied significance. These findings are important to clearly understand the complex interactions among ecosystem services at different scales and explore appropriate management methods for agricultural development and ecological conservation in agricultural basins.

The study area is described satisfactory. The research was conducted in Sihu Lake Basin (112°00′ - 114°00′E, 29°21′ - 30°00′N) (located in central Hubei Province in Central China). Unfortunately, the natural vegetation is not described. It is advisable to fix this. There is not enough information about the types of vegetation, species diversity, prevailing species, as well as the significance of it for humans.

Response 1: We fully agree with this suggestion and explained the types of vegetation, species diversity, prevailing species in SHLB. More detailed explanations have been included in Section 2.1 (Lines:112 – 114, and 119-120).

Lines: 112 – 114. “This basin has rich biological resources and biodiversity, particularly agricultural vegetation and aquatic vegetation, which have great significance for economic development and ecological conservation. With fertile land and a large network of rivers and lakes, the SHLB is an important crop production base in China, containing 10 counties and 109 administrative villages. Rice, soybean, cotton and wheat are the main types of agricultural vegetation in the SHLB; and large areas of these crops are grown in this basin, and the grain output accounts for more than 15% of the total output in Hubei Province.”

Lines 119-120: “Therefore, the effects of human activities on ecological conservation in the SHLB cannot be ignored.”

Point 2: The methodology is described in detail. The authors used modern methods of analysis. In this study, land cover was classified into seven types with a grid size of 30 m × 30 m using ArcGIS 10.2 ((i.e., agricultural land, forestland, grassland, wetland, construction land, rivers and unused land). It is necessary to explain the choice of classification. Pearson's correlation analysis was used to identify ecosystem services trade-offs and synergies at two scales in 2000, 2010 and 2020 using R software. Redundancy analysis (RDA) was used to identify the impact of landscape patterns on the ecosystem services at different periods. The detrended correspondence analysis was tested in CANOCO 4.5. The selected methods correspond to the objectives set.

Response 2: We totally agree with the reviewer’s point of view, and have added the explanation of LULC classification in Section 2.3 (Lines:127 – 129).

Lines:127 – 129. “SHLB serves critical functions in agricultural production, flood regulation and wetland protection for local areas. Different land-use types undertake different ecosystem service functions [37], and artificial channels and rivers were extensively modified to promote agricultural production in the study area. Therefore, LULC was classified into seven types with a grid size of 30 m × 30 m using ArcGIS 10.2 (i.e., agricultural land, forestland, grassland, wetland, construction land, rivers and bare land).”

Point 3: The research results are perfectly illustrated with figures and tables that are informative and do not duplicate each other. The paper contains 3 informative tables and 6 visual figures. The results are presented clearly and clearly.

Conclusions follow from the results and are reasonable.  It is desirable to expand the description of practical application for politicians and managers. How to preserve and enhance ecosystem services of different types of lands? What should you pay attention to in the first place? What difficulties can you face when organizing sustainable land use? What uncertainties stand in the way of solving problems? The article will be of interest to a wide range of readers whose scientific interests are related to ecology. Despite the fact that English is not my native language, I read the paper with interest and had no difficulties in understanding. The paper fully corresponds to the subject and level of the Land.

Response 3: This valuable suggestion has been well taken. More details, such as the implications, limitation and future work have been provided in Section 4.4 (lines: 326-342). The practical application for politicians and managers, contributions and findings have been added in the conclusion and discussion of revised manuscript (lines: 347-348, 352-353, 355-356, 360-361, 362-363).

Point 4: The paper contains many abbreviations. This complicates the perception of the results.

Response 4: We fully agree with this suggestion, the abbreviations of landscape metrics (e.g., PLANDfor, PLANDagr, PDagr, LPIfor, etc.) and Redundancy analysis (RDA) were revised in the revised manuscript.

Round 2

Reviewer 3 Report

The authors responded to all my comments and significantly improved the paper. I have no more comments.